# Production and Quality Control of [^68^Ga]Ga-FAPI-46: Development of an Investigational Medicinal Product Dossier for a Bicentric Clinical Trial

**DOI:** 10.3390/ph18101475

**Published:** 2025-09-30

**Authors:** Alessandro Cafaro, Cristina Cuni, Stefano Boschi, Elisa Landi, Giacomo Foschi, Manuela Monti, Paola Caroli, Federica Matteucci, Carla Masini, Valentina Di Iorio

**Affiliations:** 1IRCCS Istituto Romagnolo per lo Studio dei Tumori “Dino Amadori” IRST, 47014 Meldola, Italy; cristina.cuni@irst.emr.it (C.C.); giacomo.foschi@irst.emr.it (G.F.); manuela.monti@irst.emr.it (M.M.); paola.caroli@irst.emr.it (P.C.); federica.matteucci@irst.emr.it (F.M.); carla.masini@irst.emr.it (C.M.); valentina.diiorio@irst.emr.it (V.D.I.); 2Department for Life Quality Studies, University of Bologna, 47921 Rimini, Italy; stefano.boschi@unibo.it; 3Nuovo Ospedale Santo Stefano, Azienda USL Toscana Centro—Clinical Trials Task Force—Ethics and Care Unit, 59100 Prato, Italy; elisa1.landi@uslcentro.toscana.it

**Keywords:** fibroblast activation protein (FAP), [^68^Ga]Ga-FAPI-46, positron emission tomography (PET), IMPD

## Abstract

**Background/Objectives:** Fibroblast activation protein (FAP) is highly expressed in tumor stroma and selected inflammatory conditions, offering a promising target for molecular imaging. [^68^Ga]Ga-FAPI-46 is a DOTA-based FAP inhibitor with excellent tumor-to-background ratio and potential advantages over [^18^F]FDG in low-glycolytic tumors. This article aims to highlight the quality elements that are relevant to clinical practice and to describe the development of an investigational medicinal product dossier for a bicentric clinical trial involving [^68^Ga]Ga-FAPI-46. **Methods**: The radiolabeling was performed by the two facilities using different automated synthesizers, but with the same specifications and acceptance criteria **Results**: Three validation batches per site were analyzed for radiochemical/radionuclidic purity, pH, endotoxin, sterility, and bioburden according to European Pharmacopoeia standards. Stability was as sessed up to 2 h post-synthesis. All batches met predefined acceptance criteria. **Conclusions**: Despite differences in radiosynthesizer modules, product quality and process reproducibility were maintained across both centers. [^68^Ga]Ga-FAPI-46 can be reliably produced in academic settings under GMP-like conditions, enabling multicenter clinical research and facilitating IMPD submissions for broader adoption of FAP-targeted PET imaging.

## 1. Introduction

Fibroblast activation protein (FAP) is a transmembrane serine protease that belongs to the dipeptidyl peptidase 4 (DPP4) enzyme family, characterized by its endopeptidase functionality [1]. This enzyme participates in remodeling the extracellular matrix (ECM) by degrading its structural components, thereby influencing the tumor microenvironment, facilitating cellular invasion, and promoting metastasis. FAP also contributes to various pro-tumorigenic processes, including resistance to chemotherapy, induction of angiogenesis, immunomodulation, and extracellular matrix modification, all of which support tumor progression [2,3,4,5].

In healthy adult tissues, FAP expression is minimal or completely absent, typically limited to specific physiological events such as wound healing or embryonic development, where it is found in stromal fibroblasts or mesenchymal stem cells [6,7]. In contrast, FAP is markedly overexpressed in the tumor stroma of more than 90% of epithelial cancers [8,9] and is also upregulated in several chronic inflammatory conditions including hepatic fibrosis, cardiovascular pathologies, and autoimmune diseases like rheumatoid arthritis [10,11]. These expression patterns position FAP as a promising biomarker for cancer imaging and prognosis, as well as an attractive target for both diagnostic and therapeutic nuclear medicine applications [5,6,11,12,13].

The first radiolabeled FAP inhibitor (FAPI) developed was [^125^I]I-MIP-1232, a boronic-acid-containing compound evaluated in vitro [14]. However, this early agent showed suboptimal specificity and affinity for FAP when compared to other members of the DPP and prolyl oligopeptidase (PREP) families [14,15]. Subsequently, a new class of FAPI molecules was synthesized based on the *N*-(4-quinolinoyl)-Gly-(2-cyanopyrrolidine) scaffold, exhibiting significantly improved pharmacokinetics and FAP-targeting characteristics in preclinical and clinical studies [16,17].

This quinoline-based chemical platform has led to the development of several FAPI derivatives—FAPI-02, FAPI-04, FAPI-34, FAPI-46, and FAPI-74—each compatible with various radiometals, including gallium-68, fluorine-18, yttrium-90, and lutetium-177, to meet diagnostic or therapeutic requirements [18,19]. Of particular interest are derivatives equipped with a DOTA chelator (1,4,7,10-tetraazacyclododecane-N,N′,N″,N‴-tetraacetic acid), enabling efficient complexation with gallium-68 (t_½_ = 67.7 min, β^+^ = 89%, electron capture = 11%) [20]. Among these, [^68^Ga]Ga-FAPI-46 has demonstrated excellent performance in oncologic imaging, with high tumor-to-background contrast and strong correlation between tracer uptake and FAP expression [21,22,23].

Due to its high and selective uptake in malignant tissue, [^68^Ga]Ga-FAPI-46 enables non-invasive visualization of FAP activity and offers potential for patient stratification and treatment planning [21]. The radiopharmaceutical has been tested in various cancers, notably those with low [^18^F]F-FDG uptake like liver, biliary, gastrointestinal, and peritoneal tumors [24,25].

Gallium-68 PET imaging has grown in use over the past 20 years, mainly because gallium-68 is a generator-produced radionuclide with advantageous properties. Its availability from germanium-68/gallium-68 generators facilitates on-site production, making it particularly suited for early-phase clinical research and rapid clinical translation of new radiopharmaceuticals [26]. For consistent and GMP-compliant production, automated synthesis modules are typically used to reduce operator radiation exposure and standardize manufacturing processes [27].

Maintaining optimal pH during gallium-68 labeling reactions is critical and requires carefully chosen buffer systems matched to the chemical characteristics of the targeting vector [28]. Adding antioxidants like gentisic acid or ascorbic acid during synthesis improves radiochemical yields, prevents radiolysis, and helps maintain radiochemical purity (RCP) throughout the product’s shelf life [29,30].

[^68^Ga]Ga-FAPI-46 can only be used as an experimental drug in clinical trials, because there are currently no kits with market authorization and no monograph in the European Pharmacopeia (Ph. Eur.).

Our current clinical study “[^68^Ga]Ga-FAPI-46 PET/CT for Molecular Evaluation of Fibroblast Activation and Risk Stratification in Solid Tumors” (EudraCT No. 2022-003786-38) was approved by the Italian Medicines Agency (AIFA) in 2023.

This study aims to assess the diagnostic performance of [^68^Ga]Ga-FAPI-46 PET/CT in patients with solid tumors for whom conventional imaging, such as FDG-PET or morpho-functional techniques, has led to inconclusive results. The clinical protocol includes a single intravenous injection of 150–200 MBq of [^68^Ga]Ga-FAPI-46, synthesized in-house in our radiopharmacy.

For human administration, experimental radiopharmaceuticals must comply with EU Regulation 536/2014. In particular, Article 61(b) specifies that the preparation of radiopharmaceuticals used as diagnostic investigational medicinal products can be performed without a GMP certification if this process is carried out in hospitals, health centers or clinics by pharmacists or other persons legally authorized in the Member State.

In Italy, the standard framework is provided by the national “Norme di Buona Preparazione dei Radiofarmaci per Medicina Nucleare (NBP-MN), a regulatory quality system for nuclear medicine production [31].

Submission of the clinical trial to AIFA required the preparation of an Investigational Medicinal Product Dossier (IMPD) for [^68^Ga]Ga-FAPI-46 in accordance with the European Medicines Agency (EMA) guidelines [32], which define quality and safety standards for investigational medicinal products (IMPs). This paper presents the validation strategy, analytical methods, and predefined acceptance criteria for [^68^Ga]Ga-FAPI-46 to comply with the IMPD submission process.

This study is of particular interest due to the specific nature of the IMP and its reduced stability, as well as the bicentric nature of the study. The production of the IMP was performed by the two clinical centers using different equipment and procedures, but with the same acceptance criteria. For this reason, the regulatory authority required a unique IMPD for both of the clinical centers; this condition is an interesting challenge for IMPD, which we will discuss in this article.

## 2. Results

The unique IMPD for [^68^Ga]Ga-FAPI-46 has been filed following EMA guidelines [32]; this document consists of two sections: one on the drug substance and one on the investigational medicinal product.

### 2.1. Drug Substances

The IMPD for [^68^Ga]Ga-FAPI-46 describes two Drug Substances: the ligand FAPI-46 and gallium-68 in the chemical form of [^68^Ga]GaCl_3_.

#### 2.1.1. FAPI-46

Nomenclature: (*S*)-2,2′,2″-(10-(2-(4-(3-((4-((2-(2-cyano-4,4-difluoropyrrolidin-1-yl)-2-oxoethyl)carbamoyl)quinolin-6-yl)(methyl)amino)propyl)piperazin-1-yl)-2-oxoethyl)-1,4,7,10-tetraazacyclododecane-1,4,7-triyl)triacetic acid

Molecular formula: C_41_H_57_F_2_N_11_O_9_

Molecular weight: 885.96

The molecular structure of FAPI-46 is shown in Figure 1.

Detailed information for FAPI-46 can be found in the quality document “Chemistry Manufacturing and Controls” (CMC), which was provided by the producer. This document is continuously updated by the manufacturer with the latest available data. Furthermore, the results of quality controls tests for each batch of FAPI-46 are available in the Certificate of Analysis (CoA) supplied with each batch.

FAPI-46 precursor vials of 50 μg are supplied by SOFIE for our clinical trial on the basis of a Material Transfer Agreement between iTheranostics, Inc., a Delaware corporation (Dulles, VA, USA), and the two participating centers. FAPI-46 precursor vials of 50 μg are produced by ABX (Advanced Biochemical Compounds Biomedizinische Forschungsreagenzien GmbH, Radeberg, Germany), according to GMP requirements for APIs for clinical trials.

#### 2.1.2. Gallium-68

The active substance, gallium-68, is used as [^68^Ga]GaCl_3_ in a 0.1 M hydrochloric acid (HCl) aqueous solution. It is fully soluble in acidic aqueous environments, where it exists in a dissociated form.^68^GaCl_3_ → ^68^Ga^3+^ + 3Cl^−^.

Gallium-68 is obtained using germanium-68/gallium-68 generators (GalliaPharm; marketing authorization numbers 042707024 and 042707036 for Laboratory 1, and 042,707,048 for Laboratory 2, product by Eckert & Ziegler Radiopharma GmbH, Robert-Rössle-Str. 10, 13125 Berlin, Germany), following the decay of the parent radionuclide germanium-68. The generator contains germanium-68, sorbed onto a TiO_2_ column, which decays to produce the daughter nuclide gallium-68. This column is maintained in a sterile, pyrogen-free environment with 0.1 N HCl. When in use, the gallium-68 is eluted with 5 mL of sterile, apyrogenic 0.1 N HCl as [^68^Ga]GaCl_3_. This solution complies with the requirements of the Summary of Product Characteristics (SmPC) for the drug and the specific monograph of the Ph. Eur. (“Gallium (^68^Ga) Chloride Solution for Radiolabelling”, 2013:2464).

The GalliaPharm Generator is supplied with a Certificate of Conformity issued according to the specifications given in the Ph. Eur. 07/2013:2464 “Gallium (^68^Ga) Chloride Solution for Radiolabelling”, which are given below in Table 1.

Germanium-68 decays via electronic capture, with a half-life of 270.8 days. Gallium-68 then decays into its stable isotope zinc-68 through positron emission, with a maximum energy of 1.899 MeV for 88.91%, and through electronic capture for 11%. It has a half-life of 67.61 min and emits a gamma photon of 1077.3 keV (3%). The process of positron annihilation following electron emission leads to the emission of two coplanar photons of 511 keV.

The scheme for the germanium-68/gallium-68 decay is shown in Figure 2.

### 2.2. Investigational Medicinal Product Under Test (IMP)

#### 2.2.1. Description and Composition of the IMP

The IMP consists of a [^68^Ga]Ga-FAPI-46 solution with an activity range of 620–697 MBq ART for Laboratory 1 and 500–700 MBq ART for Laboratory 2 at the end of the synthesis (EOS). This is referred to as the Activity Reference Time (ART). The final volume is approximately 9 mL for Laboratory 1 (radioactive concentration 68–77 MBq/mL) and 10 ± 0.5 mL for Laboratory 2 (50–70 MBq/mL).

The IMP is formulated as a multi-dose drug with the components described in Table 2 for Laboratory 1 and Table 3 for Laboratory 2.

#### 2.2.2. Description of the Manufacturing Process and Process Controls

The radiolabeling of [^68^Ga]Ga-FAPI-46 is performed using Modular-Lab Eazy Synthesizer (Eckert & Ziegler Radiopharma GmbH, Berlin, Germany) in Laboratory 1 and Mini AllinOne-MiniAIO (Trasis SA, Ans, Belgium) in Laboratory 2.

From a chemical point of view, the reaction involves the trivalent gallium-68 ion complexing with the DOTA chelating portion, which is linked to the chemical structure of FAPI-46, as shown in Figure 1.

Figure 3 shows typical radiometric and UV chromatograms of [^68^Ga]Ga-FAPI-46 syntheses for Laboratory 1. The reaction is performed at a temperature of 95 °C. The typical synthesis time is 10 min in Laboratory 1 and 11 min in Laboratory 2.

### 2.3. Quality Controls

#### 2.3.1. Acceptance Criteria

The acceptance criteria, specifications, and release schedule were chosen in accordance with the relevant sections of the current Ph. Eur., and are reported in Table 4.

All established quality parameters must be met by the product. The dose given to each patient was between 150 and 200 MBq.

The justification for conducting post-release tests depends on the type of test and the analyte.

#### 2.3.2. Validation of the Analytical Procedures

Validation is the documented process of demonstrating that a procedure, process, equipment, material, activity, or system consistently produces the expected results. The aim is to contribute to and guarantee the quality of a radiopharmaceutical.

Validating analytical procedures demonstrates their suitability for their intended purpose.

Both laboratories carried out validation of analytical procedures, acceptance limits, and parameters (specificity, linearity, range, accuracy, precision, quantification, and detection limits) for analytical method validation according to the ICH guideline Q2(R1) [32].

For the HPLC determination of chemical purity, ^nat^Ga-FAPI-46 and FAPI-46 were used by both laboratories.

Regarding radiochemical purity, it should be noted that some validation parameters could not be quantified. Specifically, the analysis of at least two different radioactive analytes with comparable activity is required for specificity, but this was not possible for experimental reasons. When conducting analytical tests for procedure validation, the ALARA (As Low As Reasonably Achievable) rule should be followed to minimize unnecessary radioactive exposure.

The results of the validation process for the radio HPLC methods for Laboratories 1 and 2 are reported in Table 5.

Chromatograms of standard ^nat^Ga-FAPI-46 and FAPI-46 are shown in Figure 4.

#### 2.3.3. Bioburden

The product was sent for bioburden test before filtration, with 1 mL used for each test sample (Eurofins Laboratory Biolab S.r.l., Vimodrone, Milan, Italy).

The results were as follows:Total aerobic microbial count (TMAC) < 1 cfu/mL.Total yeast and mold count (TYMC) < 1 cfu/mL.where <1 cfu/mL indicates an absence of colonies.

#### 2.3.4. Batch Analysis and Process Validation

Process validation is a documented, systematic approach used to provide evidence that a manufacturing process consistently produces products that meet their predetermined specifications and quality attributes.

For process validation, three different batches of [^68^Ga]Ga-FAPI-46 were performed by both laboratories. Each production run was performed in accordance with the established validation protocol and was evaluated to confirm compliance with all specified acceptance criteria.

Due to differences in the manufacturing workflows of Laboratory 1 and Laboratory 2, each site performed its own set of three validation batches independently to ensure that the process was robust and applicable across both facilities.

The analytical parameters assessed during validation were those outlined in Table 4, and the results of three representative batches are summarized in Table 6.

All batches used for process validation complied with the acceptance criteria for both laboratories.

#### 2.3.5. Stability

Stability was assessed for two hours in both laboratories.

Three process-validation batches were stored at room temperature and tested for visual appearance, radiochemical purity (HPLC and TLC), and pH.

Table 7 shows the stability data for each laboratory.

## 3. Discussion

FAPI-46 is the result of extensive optimization and research into fibroblast activation protein (FAP) receptor ligands. Compared to alternative inhibitors, it exhibits unique chemical characteristics and, as the radiopharmaceutical [^68^Ga]Ga-FAPI-46, demonstrates superior performance in oncological imaging [33].

The compound readily incorporates the radionuclide within relatively short reaction times and at temperatures compatible with standard radiopharmaceutical equipment, typically utilizing germanium-68/gallium-68 generators.

Due to its straightforward production process and exceptional diagnostic capability, especially in cases involving tumors with low FDG uptake [34] or situations where it is unclear whether FDG uptake reflects residual tumor or an inflammatory response to treatment [35], [^68^Ga]Ga-FAPI-46 represents a valuable diagnostic option for nuclear medicine practitioners. In terms of synthesis, FAPI-46 is generally considered more robust and easier to handle than other FAP ligands such as FAPI-04. At the same time, ^18^F-labeled derivatives (e.g., [^18^F]AIF-FAPI-74) require more complex chemistry and infrastructure, but allow for large-scale production and extended distribution due to the longer half-life of fluorine-18 [36].

Regarding stability, our data demonstrate that [^68^Ga]Ga-FAPI-46 maintains a radiochemical purity above 95% up to 2 h post-synthesis, which is adequate for routine clinical application and dose fractionation. Other groups have reported strategies to prolong stability, such as optimized formulations or the use of cyclotron-produced gallium, whereas ^18^F-labeled analogues display stability profiles dependent on the labeling method and formulation buffer. Overall, [^68^Ga]Ga-FAPI-46 offers a reliable balance between ease of production and sufficient shelf-life for clinical PET imaging, while ^18^F-labeled variants are more suitable for centralized production and delayed imaging protocols [25,29].

In terms of diagnostic performance, several comparative studies indicate that FAPI-46 provides excellent tumor uptake and tumor-to-background ratios, often outperforming FAPI-04 in clinical series. However, modified derivatives, such as dimeric FAPIs, have been shown to improve tumor retention and may enhance sensitivity in certain oncological settings. Likewise, ^18^F-FAPI tracers offer practical advantages for delayed imaging and multicentric studies, but their diagnostic accuracy has not consistently surpassed that of [^68^Ga]Ga-FAPI-46. Taken together, these findings support the use of [^68^Ga]Ga-FAPI-46 as a practical and reliable PET tracer, with the choice between different FAPI derivatives depending on clinical requirements, production facilities, and study design [20,37]

Currently, the Ph. Eur. does not include a monograph for the radiopharmaceutical [^68^Ga]Ga-FAPI-46. No industrially produced kits with marketing authorization are available for extemporaneous labeling at nuclear medicine radiopharmacies. Consequently, the radiopharmaceutical can be used only in clinical studies as an experimental radiopharmaceutical.

The EU Regulation 536/2014 gives an achievable opportunity for academic nuclear medicine to promote non-profit clinical trials using diagnostic experimental compounds [^68^Ga]Ga-FAPI-46. Article 61(b), in particular, specifies that the preparation of radiopharmaceuticals used as investigational medicinal products for diagnosis, where this process is carried out in hospitals, health centers, or clinics by pharmacists or other persons legally authorized in the Member State, can be performed without GMP certification.

According to applicable European regulations, obtaining authorization for a clinical trial necessitates submission of a study protocol to regulatory authorities, accompanied by an IMPD prepared in accordance with EMA guidelines.

Moreover it is now also possible to design academic multicentric clinical trials with [^68^Ga]Ga-FAPI-46 and in general with experimental radiopharmaceuticals labeled by Ga-68, preparing an unique IMPD with different sections for radiolabeling but with the same acceptance criteria. This favorable scenario enables clinical centers with different equipment to be networked, referring to the same acceptance criteria.

The outcomes described in the preceding section confirm that different production methods do not compromise the finished product’s quality or stability, referring to the same acceptance criteria.

Overall, these findings indicate that the manufacture and quality assurance of [^68^Ga]Ga-FAPI-46 are both practical and reliable. Despite procedural differences between centers, the simplicity of its production and quality control supports broader application for PET imaging with [^68^Ga]Ga-FAPI-46 and the establishment of compliant protocols analogous to those implemented in our clinical trial.

## 4. Materials and Methods

### 4.1. Description of [^68^Ga]Ga-FAPI-46 Manufacturing Process

#### 4.1.1. Set up of Radiosynthesizer

For Laboratory 1, the radiosyntheses were performed by Modular-lab Eazy (Eckert & Ziegler Eurotope Gmbh—Robert Rössle-Straße 10, Berlin, Germany), without the use of organic solvents. The module was placed in a shielded isolator (Elena Beta, COMECER S.p.a., Castelbolognese, Italy) which was situated within a class D Laboratory.

Figure 5a shows the user graphic interface of the synthesis process.

The software, cassettes, reagent set, and detailed instructions for radiosynthesis were supplied by Eckert and Ziegler Eurotope Gmbh (Robert Rössle-Straße 10, Berlin, Germany).

For Laboratory 2, the radiosyntheses were performed by MiniAllinOne—MIiniAIO Trasis (TRASIS SA. Rue Gilles Magnée 90, 4430 Ans, Belgium).

The module was placed in a shielded isolator (Hot cell H700, TRASIS Rue Gilles Magnée 90, 4430 Ans, Belgium), which was situated within a Class D laboratory.

Figure 5b shows the user graphic interface of the synthesis process.

The software, cassette, reagents set, and detailed instructions for radiosynthesis of [^68^Ga]Ga-FAPI-46 were provided by Trasis (Rue Gilles Magnée 90, 4430 Ans, Belgium)

#### 4.1.2. Reagents

For Laboratory 1, the reagent set included the following:Reagent set product by ABX (Heinrich-Glaeser- Straße 10-14, 01454 Radeberg, Germany), composed of the following:
○Vial 1 (EZ-102-V1) containing 5 mL of NaCl 5 M/HCl 30%;○Vial 2 (EZ-102-V2) containing 680 mg of sodium acetate trihydrate;○Vial 5 (EZ-102-V5) containing 3 mg of ascorbic acid.Water for injectable preparations (100 mL bottles) with MA was purchased from Monico S.p.A., Venezia/Mestre, Italy.Sodium chloride 0,9% 100 mL with an MA was purchased from Fresenius Kabi S.r.l., Isola della Scala, Italy.Single-use sterile cassettes were produced by Eckert & Ziegler Eurotope GmbH (Robert Rossle- Straße 10, Berlin, Germany) and provided by Radius (RADIUS s.r.l. Via Luigi Menarini, 31, 40054 Budrio, BO, Italy).Sterilizing filter 0.22 mm (product code: SY25PL-S-MDI Advanced Microdevices PVT LTD 21 Ind., Area Ambala Canti 133006, India).

For Laboratory 2, the reagent set included the following:Reagent set produced by Trasis (Rue Gilles Magnée 90, 4430 Ans, Belgium), composed of the following:
○Part 1: Syringe containing E&Z Eluent; syringe containing acetate buffer; ethanol vial; and sodium Chloride 0.9% (BBraun-Melsungen AG, 34209 Melsungen, Germany).○Part 2: Sodium ascorbate.Single-use sterile cassette product by Medline Liége Science Park-Rue des Gardes-Frontiére 5, 4031 Angleur Belgium, distributed by Trasis (Rue Gilles Magnée 90, 4430 Ans, Belgium). The cassette includes a solid phase extraction (SPE) cartridge (Oasis HLB Plus Short Cartridge, 225 mg sorbent per cartridge, 60 mm, 50/pk).Sterilizing filter 0.22 mm (product code: 6,764,192 PALL Medical Avenue de Tivoli 3, CH-1700 Fribourg, Switzerland).

#### 4.1.3. Process Description

The synthesis process used by Laboratory 1 was developed by Eckert & Ziegler Eurotope GmbH (Robert-Rossle-Straße 10, Berlin, Germany) in collaboration with ABX Advanced Biochemical Compounds Biomedizinische Forschungsreagenzien GmbH (Heinrich-Gleaser-Straße 10–14, 01454 Radeberg, Germany).

The first step of the radiosynthesis involves eluting [^68^Ga]GaCl_3_ from the generator germanium-68/gallium-68 and trapping it inside a pre-purification cartridge containing a strong cation exchange resin, which is included in the previously described cassette used at Laboratory 1. Gallium-68 is then eluted with 1.1 mL of an acidified NaCl solution (15 µL of HCl added to NaCl solution) and transferred to a reaction vial containing 400 µL of acetate buffer solution at pH 4.5, 50 µg of the FAPI-46 precursor, and 3 mg of ascorbic acid (added as a scavenger). After an 11 min incubation period at 95 °C, the reaction mixture is diluted with 7.5 mL of 0.9% sodium chloride solution and purified using a cation exchange resin to retain any free ^68^Ga^3+^. The mixture is then filtered terminally through a 0.22 µm sterilizing filter.

Similarly to the process described for Laboratory 1, the first step in Laboratory 2 also involves eluting [^68^Ga]GaCl_3_ from the germanium-68/gallium-68 generator using 5 mL of 0.1 M HCl (syringe containing E&Z eluent, kit part 1). The eluate is transferred directly to the reactor without pre-purification. The FAPI-46 precursor (50 µg) has already been transferred to the reactor and preliminarily solubilized in 1 mL of acetate buffer (syringe containing acetate buffer, kit part 1). To this, 0.5 mL of ascorbate buffer is added, giving a final pH of 4.0. The reaction mixture is then heated to 95 °C for 10 min. The reaction mixture is diluted with 5 mL of ascorbate buffer to cool it and is then passed through an OASIS HLB Plus short cartridge, which is capable of selectively retaining [^68^Ga]Ga-FAPI-46. The reactor is then washed with 5 mL of ascorbate buffer to remove any product residues. The purified product is eluted from the SPE cartridge using 0.7 mL of ethanol and formulated with 9.5 mL of ascorbate buffer. The resulting [^68^Ga]Ga-FAPI-46 solution is sterilized by filtration using a 0.22 µm filter placed on the final vial and is then used for quality control and subsequent fractionation into doses.

The product vial activity is measured in a dose calibrator at the end of the synthesis to verify the expected activity and calculate the activity concentration.

### 4.2. Quality Control

#### 4.2.1. Standard Procedures

The pH value of the formulation was determined using pH strips (Merck pH indicator strip, Acilit, increment 0,5 pH unit).The endotoxin test was performed using the Limulus Amebocyte Lysate test (LAL test), on an Endosafe Nexgen-PTSTM (Charles River Laboratories, 26866 Sant’Angelo Lodigiano (LO), Italy).As required by national quality assurance regulations (NBP MN), since this preparation cannot proceed to terminal sterilization, it must be sterilized by filtration using a sterile disposable membrane with 0.22 µm pores. Filter integrity must be checked by the bubble point test before release:
○For Laboratory 1, the bubble point test was performed on an Integritest 4 system (Merck Millipore KGaA, Darmstadt, Germany).○For Laboratory 2, the bubble point test was performed automatically by the synthesis module (Mini All-in-One Trasis—Hot Cell H700).The sterility test was performed by an external laboratory for both Laboratory 1 and Laboratory 2.Radionuclidic purity was assessed in both laboratories by measuring the half-life and identifying characteristic emission peaks for gallium-68, in accordance with the current Ph. Eur. Monograph 2.2.66 “Detection and Measurement of Radioactivity.”

#### 4.2.2. HPLC Analysis

Both laboratories purchased standard natGa-FAPI-46 and FAPI-46 from ABX GmbH—Advanced Biochemical Compounds (Radeberg, Germany).

Laboratory 1 performed HPLC analysis on an Ultimate 3000 system equipped with UV variable wavelength detector RS300 (Thermo Fischer Scientific, Dreieich, Germany) and a radiometric detector (GABI, Raytest, Straubenhardt, Germany). The system was controlled using Chromeleon software version 7.2 SR5 (Dionex, Sunnyvale, CA, USA).

For Laboratory 2, HPLC analysis was performed on a RadioHPLC (ITG) equipped with UV detector S3245 and a radiometric detector in series S 3700 (ITG)

Both laboratories used the following column: Acclaim 120 C18, 3 µm, 120 Å, 3 × 150 mm (Thermo Scientific, Waltham, MA, USA).

Both laboratories performed HPLC analysis using a multi-step gradient with solvent A (0.1% trifluoroacetic acid, TFA, in water) and solvent B (0.1% TFA in acetonitrile). The method involves a 10 min gradient elution from 95% A and 5% B to 50% A/B, maintained for two minutes. This is followed by a 4 min gradient return to the initial conditions, which are then held until the end of the run, making a total run time of 20 min. The flow rate was set to 0,6 mL/min, UV wavelength was set to 205 nm and column oven was set to 25 °C. The injection volume was 20 µL.

#### 4.2.3. Thin Layer Chromatography (TLC)

TLC was performed using ITLC-SG chromatography paper Agilent (5301 Stevens Creek Blvd., Santa Clara, CA 95051, USA). Approximately 1–2 µL of the IMP injection solution was spotted onto the plate. The solvent used to develop the TLC plates was a 50:50 *v/v* mixture of ammonium acetate (0.1 N) and methanol. The development plate was analyzed using autoradiography on an MS (multi-sensitive) storage phosphor screen and by a Cyclone Plus Storage Phosphor system (Perkin Elmer, Waltham, MA, USA) in Laboratory 1. A TLC ScanRam (Lab Logic, Sheffield, UK) radiodetector that does not require the exposure of a radiosensitive plate but reads the plate directly was used in Laboratory 2.

## 5. Conclusions

This study demonstrates that [^68^Ga]Ga-FAPI-46 can be produced in compliance with regulatory requirements by using different production and quality control methods at each site, with a unified IMPD covering all EMA guidelines. Although the two centers use distinct synthesis methods, no differences are reported in the final products, quality, or stability.

Overall, the results show that the production and quality control of [^68^Ga]Ga-FAPI-46 are both practicable and dependable. The ease of production and quality control enables a broader applicability for nuclear medicines and radiopharmacies with experience in gallium-labeled compounds, allowing the implementation of an academic network based on the same quality criteria.

## Figures and Tables

**Figure 1 pharmaceuticals-18-01475-f001:**
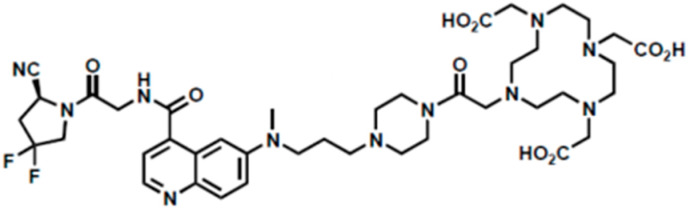
The chemical structure of FAPI-46.

**Figure 2 pharmaceuticals-18-01475-f002:**
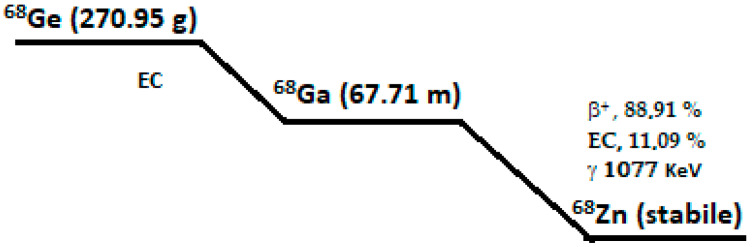
Gallium-68 decay scheme.

**Figure 3 pharmaceuticals-18-01475-f003:**
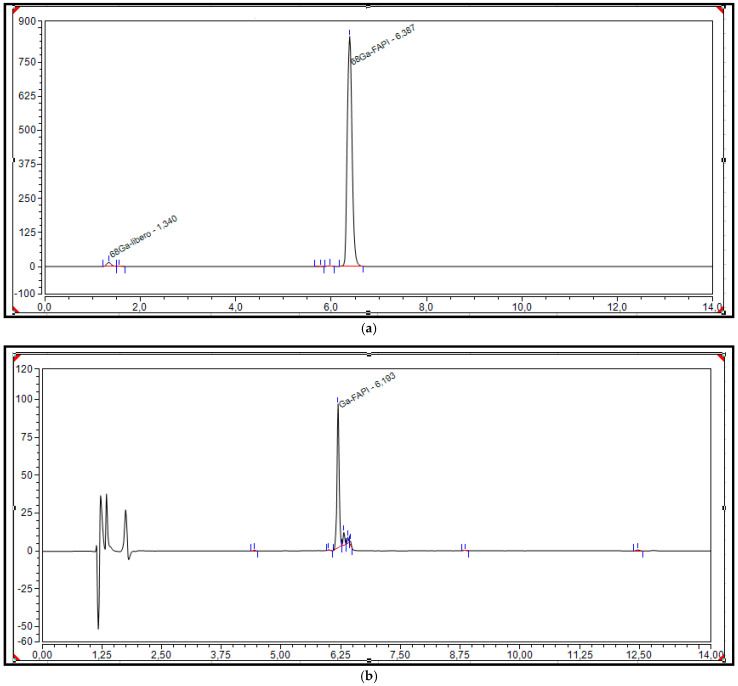
Relevant chromatograms of the final products of [^68^Ga]Ga-FAPI-46 with radiometric (**a**) and UV detectors (**b**).

**Figure 4 pharmaceuticals-18-01475-f004:**
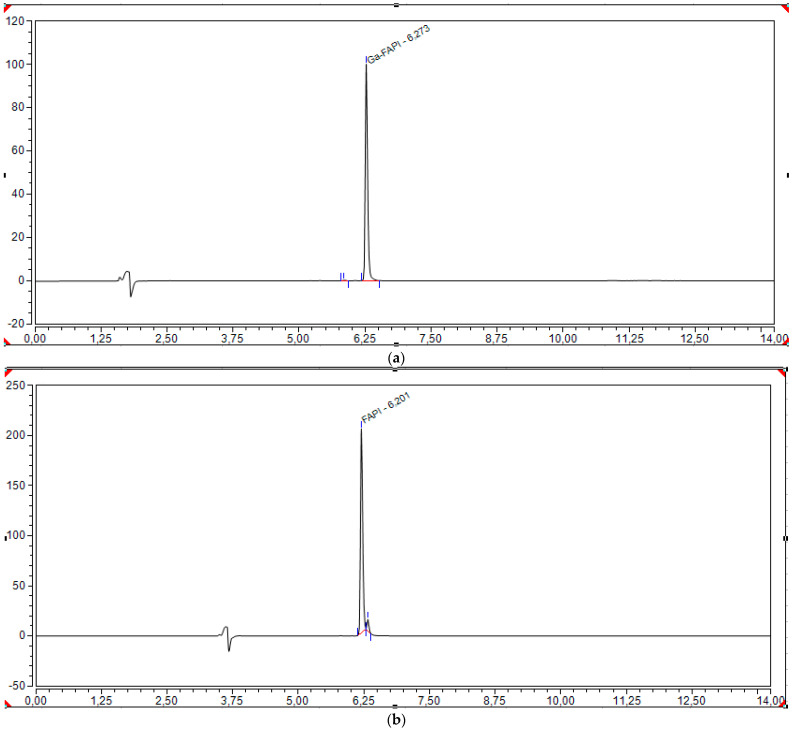
Chromatograms of standard solutions of (**a**) ^nat^Ga-FAPI-46 (20 µL of a 0.1 mg/mL solution and (**b**) FAPI-46 (20 µL of a 0.04 mg/mL solution. Rt of ^nat^Ga-FAPI-46 is slightly lower than that of [^68^Ga]Ga-FAPI-46 with radiometric detection because it is positioned after the UV detector.

**Figure 5 pharmaceuticals-18-01475-f005:**
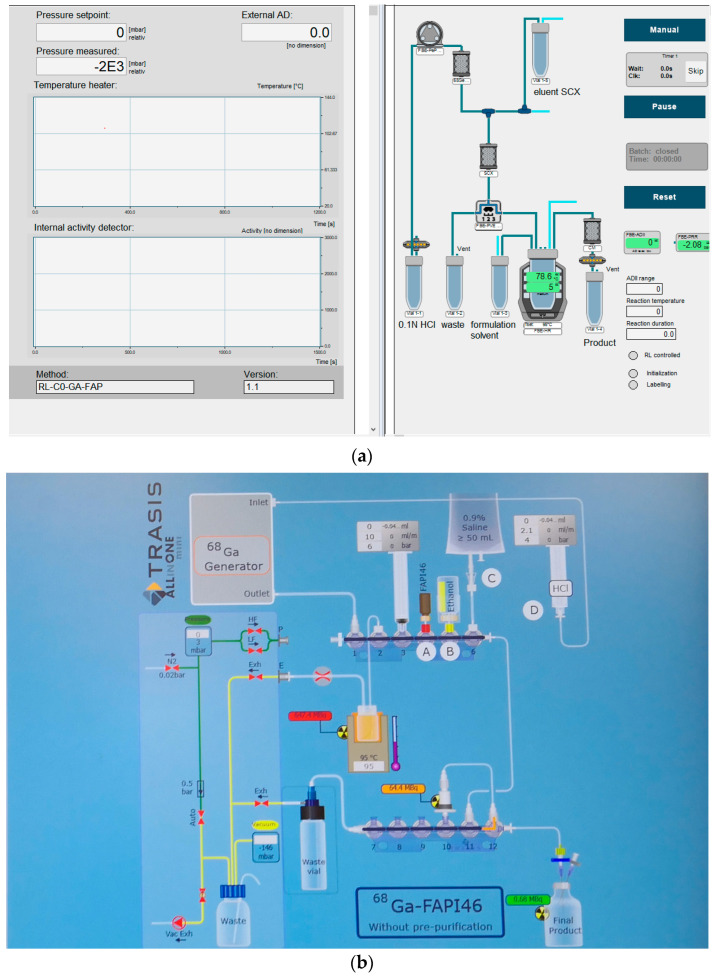
User graphic interface of the synthesis cassette for the preparation of [^68^Ga]Ga-FAPI-46 for Laboratory 1 (**a**) and for Laboratory 2 (**b**).

**Table 1 pharmaceuticals-18-01475-t001:** “Gallium (^68^Ga) Chloride Solution for Radiolabelling” specifications.

Parameters	Acceptance Criteria
Appearance	Clear and colorless solution
Radionuclidic purity	≥99.9%
^68^Ge breakthrough	≤0.001%
Non-radioactive metals (ICP-EOS)	Iron: <10 μg/GBq Zinc: <10 μg/GBq
Identity	Gamma spectrometry: 0.511; 1.077 MeV (a sum peak may be observed at 1.022 MeV)
Half-life	62–74 min
Radiochemical purity	ITLC-SG; mobile phase: methanol/ammonium acetate (1:1) ≥ 95%
pH	≤2
Endotoxin level	≤175 EU/V

**Table 2 pharmaceuticals-18-01475-t002:** Batch formula of [^68^Ga]Ga-FAPI-46 for Laboratory 1.

Components	Function	Amount/Activity
[^68^Ga]Ga-FAPI-46	Active Pharmaceutical Ingredient	620–697 MBq ART
Sodium Chloride NaCl ≥ 99.99% Suprapur	Eluent	312.4 mg
Chloridric Acid HCl 30% TraceSELECT Ultra		15 µL
Ultrapure Water TraceSELECT Ultra, ACS Reagent		1.08 mL
Chloridric Acid HCl 30% TraceSELECT Ultra	Reaction Buffer	8.8 µL
Acetic Acid ≥ 99.5%		20 µL
Sodium Acetate Trihydrate BioUltra ≥ 99.5%		60.4 mg
Ultrapure Water TraceSELECT Ultra, ACS Reagent		0.37 mL
Ascorbic Acid	Radical Scavenger	0.3 mg
Sodium Chloride 0.9%	Diluent/Excipient	7.5 mL

**Table 3 pharmaceuticals-18-01475-t003:** Batch formula of [^68^Ga]Ga-FAPI-46 for Laboratory 2.

Components	Function	Amount/Activity
[^68^Ga]Ga-FAPI-46	Active Pharmaceutical Ingredient	500–700 MBq ART
Ethanol Absolute 100%, EMSURE ACS, ISO, Eur Ph Reag.	Eluent	0.7 mL
Sodium Ascorbate Ph Eur	Radical Scavenger	0.1 g
Sodium Chloride 0.9%	Diluent/Excipient	9.5 mL

**Table 4 pharmaceuticals-18-01475-t004:** Recommended tests for quality controls and pre-/post-release time schedule.

Parameter	Method	Acceptance Criteria	Pre/Post Release
[^68^Ga]Ga-FAPI-46 activity	Dose calibrator	Lab 1: 620–697 MBq Lab 2: 500–700 MBq	Pre
Radioactive concentration	Dose calibrator	Lab 1: 68–78 MBq/mL Lab 2: 50–70 MBq/ml	Pre
Appearance	Visual inspection	Clear and colorless solution	Pre
Identification	γ-spectrometry	Peaks at 0.511 and 1022 Mev	Pre
	Half-life	62–74 min.	
Identification	HPLC	T_R_ [^68^Ga]Ga-FAPI-46 ± 0.2 min T_R_ ^nat^Ga-FAPI-46	Pre
Radiochemical purity	TLC	[^68^Ga]Ga-FAPI-46 ≥ 95%—[^68^Ga]Ga^3+^ ≤ 3%	Pre
Radiochemical purity	HPLC	[^68^Ga]Ga-FAPI-46 ≥ 95%—[^68^Ga]Ga^3+^ and other radiolysis products ≤ 5% of which [^68^Ga]Ga^3+^ ≤ 2%	Pre
System suitability	HPLC	Symmetry factor [^68^Ga]Ga-FAPI-46 ≤ 2.5	Pre
pH	pH strips	4.0–8.0	Pre
Filter integrity	Bubble point test	≥50 psi	Pre
Radionuclidic purity	γ-spectrometry	≤0.001%	Post
Sterility	Sterility test (Ph. Eur.)	Sterile	Post
Bacterial endotoxin	Ph. Eur.	≤175 EU/V	Pre

**Table 5 pharmaceuticals-18-01475-t005:** Parameters and acceptance criteria for the validation of the radio-HPLC method and the obtained results.

Chemical Purity UV Detector			
Parameters	Acceptance Criteria	Lab 1 Results	Lab 2 Results
Specificity	Rs ^nat^Ga-FAPI-46 and FAPI-46 Rs ≥ 1.5	Comply	Comply
Precision	CV% FAPI-46 ≤ 5% CV% ^nat^Ga-FAPI-46 ≤ 5%	2% 2.6%	2.3% 2.4%
Linearity	R^2^ FAPI-46 ≥ 0.99 R^2 nat^Ga-FAPI-46 ≥ 0.99	0.999 0.998	0.998 0.999
Limit of quantification LOQ (μg/mL)	Experimental	FAPI-46 = 0.39 ^nat^Ga-FAPI-46 = 0.74	FAPI-46 = 1.91 ^nat^Ga-FAPI-46 = 0.58
Limit of detection LOD (μg/mL)	Experimental	FAPI-46 = 0.13 ^nat^Ga-FAPI-46 = 0.39	FAPI-46 = 0.63 ^nat^Ga-FAPI-46 = 0.19
Range accuracy	Average bias < 5%	Comply	Comply
**Radiochemical Purity Radiodetector**			
**Parameters**	**Acceptance Criteria**	**Lab 1 Results**	**Lab 2 Results**
Specificity	Not applicable	NA	NA
Precision	CV% ≤ 5%	3.1%	3.1%
Linearity	R^2^ ≥ 0.99	0.9983	0.992
Limit of quantification LOQ (MBq/mL)	Experimental	30.01	4.5
Limit of detection LOD (MBq/mL)	Experimental	9.9	1.5
Range accuracy	Average bias < 5%	Comply	Comply

NA: not applicable.

**Table 6 pharmaceuticals-18-01475-t006:** Results of [^68^Ga]Ga-FAPI-46 representative batches.

Parameters	Method	Acceptance Criteria	Laboratory 1			Laboratory 2		
			Batch 1 8 Apr. 2022	Batch 2 12 May 2022	Batch 3 18 May 2022	Batch 1 8 Aug. 2023	Batch 2 10 Aug. 2023	Batch 3 11 Aug. 2023
[^68^Ga]Ga-FAPI-46 activity	Dose calibrator	Lab 1: 620–697 MBq Lab 2: 500–700 MBq	628 MBq	662 MBq	697 MBq	644 MBq	662 MBq	593 MBq
Radioactive concentration	Dose calibrator	Lab 1: 68–78 MBq Lab 2:50–70 MBq	69.7 MBq/mL	73.5 MBq/mL	77.4 MBq/mL	64.4 MBq/mL	66.2 MBq/mL	59.3 MBq/mL
Volume	-	Lab 1: 9 mL Lab 2: 10 ml	Comply	Comply	Comply	Comply	Comply	Comply
Appearance	Visual test	Clear and colorless solution	Comply	Comply	Comply	Comply	Comply	Comply
Identification	HPLC	T_R_ [^68^Ga]Ga-FAPI-46 ± 0.2 min T_R_ ^nat^Ga-FAPI-46	0.157 min	0.172 min	0.144 min	0.141 min	0.160 min	0.156 min
Radionuclidic identity	γ-spectrometry	Peaks at 0.511 and 1022 Mev	Comply	Comply	Comply	Comply	Comply	Comply
	Half-life	62–74 min.	67.85 min	69.39 min	67.83 min	68.2 min	68.9 min.	66.9 min.
Radiochemical purity	TLC	[^68^Ga]Ga-FAPI-46 ≥ 95%—[^68^Ga]Ga^3+^ ≤ 3%	99.6%	99.9%	99.2%	99.2%	99.7%	99.6%
			0.4%	0.1%	0.8%	0.8%	0.3%	0.4%
Radiochemical purity	HPLC	[^68^Ga]Ga-FAPI-46 ≥ 95%	98.1%	99.0%	97.1%	99.9%	99.7%	99.9%
		[^68^Ga]Ga^3+^ and other radiolysis products ≤ 5%	1.9%	1.0%	2.9%	0.1%	0.3%	0.0%
		[^68^Ga]Ga^3+^ ≤ 2%	1.6%	0.7%	2.0%	0.1%	0.3%	0.0%
System suitability	HPLC	Symmetry factor [^68^Ga]Ga-FAPI-46 ≤ 2.5%	Comply	Comply	Comply	Comply	Comply	Comply
pH	pH strips	4.0–8.0	4.5	4.5	4.5	5.8	6.0	5.9
Filter integrity	Bubble point test	≥50 psi	≥50 psi	≥50 psi	≥50 psi	≥50 psi	≥50 psi	≥50 psi
Radionuclidic purity	γ-spectrometry	≤0.001%	0.000035%	0.000018%	0.000034%	<2 × 10^−5^%	<2 × 10^−5^%	<2 × 10^−5^%
Sterility	Sterility test (Ph. Eur.)	Sterile	Comply	Comply	Comply	Comply	Comply	Comply
Bacterial endotoxin	Ph. Eur.	≤175 EU/V	Comply	Comply	Comply	Comply	Comply	Comply

**Table 7 pharmaceuticals-18-01475-t007:** Stability test of injectable solution of [^68^Ga]Ga-FAPI-46.

1 h Stability Test								
Parameters	Method	Acceptance Criteria	Laboratory 1			Laboratory 2		
			Batch 1 8 Apr. 2022	Batch 2 12 May 2022	Batch 3 18 May 2022	Batch 1 8 Aug. 2023	Batch 2 10 Aug. 2023	Batch 3 11 Aug. 2023
Appearance	Visual test	Clear and colorless solution	Comply	Comply	Comply	Comply	Comply	Comply
Radiochemical purity	TLC	[^68^Ga]Ga-FAPI-46 ≥ 95%—[^68^Ga]Ga^3+^ ≤ 5%	99.6%	99.9%	99.6%	98.9%	99.6%	99.5%
			0.4%	0.1%	0.4%	1.1%	0.4%	0.5%
Radiochemical purity	HPLC	[^68^Ga]Ga-FAPI-46 ≥ 95%	98.0%	98.7%	97.0%	99.8%	99.7%	99.8%
		[^68^Ga]Ga^3+^ and other radiolysis products ≤ 5%	2.0%	1.3%	3.0%	0.2%	0.3%	0.2%
		[^68^Ga]Ga^3+^ ≤ 2%	1.7%	0.8%	2.0%	0.1%	0.3%	0.1%
pH	pH strips	4.0–8.0	4.5	4.5	4.5	5.8	5.9	5.9
**2 h Stability Test**								
**Parameters**	**Method**	**Acceptance Criteria**	**Laboratory 1**			**Laboratory 2**		
			**Batch 1** **8 Apr. 2022**	**Batch 2** **12 May 2022**	**Batch 3** **18 May 2022**	**Batch 1** **8 Aug. 2023**	**Batch 2** **10 Aug. 2023**	**Batch 3** **11 Aug. 2023**
Appearance	Visual test	Clear and colorless solution	Comply	Comply	Comply	Comply	Comply	Comply
Radiochemical purity	TLC	[^68^Ga]Ga-FAPI-46 ≥ 95%—[^68^Ga]Ga^3+^ ≤ 5%	99.5%	99.7%	99.5%	98.6%	99.5%	99.3%
			0.5%	0.3%	0.5%	1.3%	0.5%	0.6%
Radiochemical purity	HPLC	[^68^Ga]Ga-FAPI-46 ≥ 95%	97.5%	98.5%	96.7%	99.3%	98.5%	99.3%
		[^68^Ga]Ga^3+^ and other radiolysis products ≤ 5%	2.5%	1.5%	3.3%	0.7%	0.6%	0.7%
		[^68^Ga]Ga^3+^ ≤ 2%	1.7%	0.8%	2.0%	0.6%	0.1%	0.1%
pH	pH strips	4.0–8.0	4.5	4.5	4.5	5.8	5.8	5.9

## Data Availability

Data presented in this study is contained within the article. Further inquiries can be directed to the corresponding author.

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
