# Peer review of "Production and Quality Control of [68Ga]Ga-FAPI-46: Development of an Investigational Medicinal Product Dossier for a Bicentric Clinical Trial"

_pharmaceuticals, 2025, doi:10.3390/ph18101475_

Round 1
Reviewer 1 Report
Comments and Suggestions for Authors
The research presented in this paper seeks to highlight the quality aspects that could influence clinical practice, based on the preparation of an investigational new drug application for a bicentric clinical trial with [68Ga]Ga-FAPI-46. However, the manuscript has a significant limitation in terms of originality, as there are multiple previous publications reporting the production of this radiopharmaceutical under very similar conditions, as evidenced in the following studies:
- Zanoni L, Fortunati E, Cuzzani G, Malizia C, Lodi F, Cabitza VS, Brusa I, Emiliani S, Assenza M, Antonacci F, Giunchi F, Degiovanni A, Ferrari M, Natali F, Galasso T, Bandelli GP, Civollani S, Candoli P, D'Errico A, Solli P, Fanti S, Nanni C. [68Ga]Ga-FAPI-46 PET/CT for Staging Suspected/Confirmed Lung Cancer: Results on the Surgical Cohort Within a Monocentric Prospective Trial. Pharmaceuticals (Basel). 2024 Nov 1;17(11):1468. doi: 10.3390/ph17111468. PMID: 39598380; PMCID: PMC11597145.
- aty LP, Degueldre S, Provost C, Schmitt C, Trump L, Fouque J, Vriamont C, Valla F, Gendron T, Madar O. Development of a versatile [68Ga]Ga-FAPI-46 automated synthesis suitable to multi-elutions of germanium-68/gallium-68 generators. Front Chem. 2024 Jul 15;12:1411312. doi: 10.3389/fchem.2024.1411312. PMID: 39076612; PMCID: PMC11284080.
- Rosenberg AJ, Cheung YY, Liu F, Sollert C, Peterson TE, Kropski JA. Fully automated radiosynthesis of [68Ga]Ga-FAPI-46 with cyclotron produced gallium. EJNMMI Radiopharm Chem. 2023 Oct 16;8(1):29. doi: 10.1186/s41181-023-00216-0. PMID: 37843670; PMCID: PMC10579206.
- Alfteimi A, Lützen U, Helm A, Jüptner M, Marx M, Zhao Y, Zuhayra M. Automated synthesis of [68Ga]Ga-FAPI-46 without pre-purification of the generator eluate on three common synthesis modules and two generator types. EJNMMI Radiopharm Chem. 2022 Jul 29;7(1):20. doi: 10.1186/s41181-022-00172-1. PMID: 35904684; PMCID: PMC9338183.
- Mallapura H, Ovdiichuk O, Jussing E, Thuy TA, Piatkowski C, Tanguy L, Collet-Defossez C, Långström B, Halldin C, Nag S. Microfluidic-based production of [68Ga]Ga-FAPI-46 and [68Ga]Ga-DOTA-TOC using the cassette-based iMiDEV™ microfluidic radiosynthesizer. EJNMMI Radiopharm Chem. 2023 Dec 13;8(1):42. doi: 10.1186/s41181-023-00229-9. PMID: 38091157; PMCID: PMC10719436.
- Spreckelmeyer S, Balzer M, Poetzsch S, Brenner W. Fully-automated production of [68Ga]Ga-FAPI-46 for clinical application. EJNMMI Radiopharm Chem. 2020 Dec 17;5(1):31. doi: 10.1186/s41181-020-00112-x. PMID: 33331982; PMCID: PMC7746794.
On the other hand, the sections indicate that validation was carried out in two different laboratories, showing different methodologies for quality control testing and compliance with specifications. However, only six batches were manufactured in total (three per laboratory) and the number of samples analyzed per batch is not specified. Everything seems to indicate that only one sample per batch was evaluated, which is not a significant size for applying a robust statistical analysis or for supporting the claim that the process is validated.
The images presented are of low quality and, in some cases, are only screenshots of the equipment. In addition, the discussion of the results is limited and insufficient.
Author Response
Dear Reviewer,
I hope that the following clarifications will better explain why we believe this article may be useful for many researchers who may wish to replicate what has been done by our group in the future.
The publications you cited, although all related to the use and production of the same radiopharmaceutical, are focused on different aspects.
Specifically, the publication by Zanoni L. et al. mainly reports clinical data concerning the diagnostic performance of the radiopharmaceutical and is centered on aspects not considered in our initial manuscript submission. We plan to address these aspects in a future publication, in which we will take into account the results of the clinical study entitled “[68Ga]Ga-FAPI-46 PET/CT for Molecular Evaluation of Fibroblast Activation and Risk Stratification in Solid Tumors”, mentioned in the manuscript. However, following your comment and that of another reviewer, we decided to expand the discussion section by including some elements related to diagnostic performance, even though this is not the main focus of our article, as we recognize it may be of interest to readers approaching the topic for the first time.
The other cited articles, in our opinion, are focused on different aspects than those addressed in our work, specifically on the optimization of the production process. In our manuscript, we clearly stated that the radiosynthesis instructions were taken from Eckert & Ziegler for Laboratory 1 and from Trasis for Laboratory 2 (lines 345 and 253), and that the study and optimization of the process were not objectives of our publication.
As underlined in lines 96 to 115, the aim of our paper is to describe the peculiarities related to the submission of an Investigational Medicinal Product Dossier (IMPD) for a diagnostic radiopharmaceutical. Due to stability issues, the competent regulatory agency required us to submit a single IMPD covering the different production and quality control aspects between the two laboratories participating in the same clinical trial. This approach to IMPD development is not only relevant for the radiopharmaceutical discussed in our article but may also serve as a general model applicable to any other diagnostic radiopharmaceutical used in multicenter clinical trials. For isotopes with short half-life and limited stability, local production at the clinical site is often required. For centers wishing to work within a network on multicenter studies, understanding how to prepare an IMPD—something that was unknown to us before undertaking this work—may facilitate rapid patient recruitment across different sites, thereby accelerating the collection of diagnostic performance data needed for future clinical practice use of the radiopharmaceutical. We are aware that this “model” may not be of interest in all countries (for instance, it is not required in Germany). However, most European regulatory agencies do require a process such as the one described in our work.
With regard to the samples analyzed per batch, we confirm that only one sample per batch was analyzed, in compliance with the EMA guidelines cited in the article (Requirements to the chemical and pharmaceutical quality documentation concerning investigational medicinal products in clinical trials – Scientific guideline, available at: https://www.ema.europa.eu/en/requirements-chemical-pharmaceutical-quality-documentation-concerning-investigational-medicinal-products-clinical-trials-scientific-guideline), and this was approved by the competent regulatory authority.
The discussion section has been revised as suggested. In particular, following the recommendation of another reviewer, we added a brief comparative discussion of the molecule studied in our manuscript with other FAPI derivatives in terms of chemical synthesis, stability, and diagnostic performance. Any further suggestions you may have will be extremely valuable to us, and we remain open to considering them for additional improvements.
The quality of the images has been improved as requested.
We hope that with these clarifications, the value of our work is now clearer.
Thank you for the time you dedicated to reviewing our manuscript.
Reviewer 2 Report
Comments and Suggestions for Authors
The authors have conducted an interesting study on the production of [^68Ga]Ga-FAPI-46 using an automated flow system. The practical production method, along with a consistent quality control process, can support broader applications in radiopharmaceuticals. My comments on the figure and the additional chemical structure have been incorporated into the attached reviewed file

Author Response
Dear Reviewer,
We are very grateful for your positive feedback on our work, and we confirm that we have carefully considered your corrections and implemented the suggested changes in the manuscript.
Thank you for the time you dedicated to reviewing our work.
Reviewer 3 Report
Comments and Suggestions for Authors
The authors presented a manuscript on “Production and Quality Control of [⁶⁸Ga]Ga-FAPI-46: Development of an Investigational Medicinal Product Dossier for a Bicentric Clinical Trial.”
Fibroblast activation protein (FAP) is a well-established target. It is a unique cell surface antigen associated with granulation tissue involved in wound healing, fetal mesenchymal fibroblasts, and the reactive stromal fibroblasts surrounding epithelial malignancies. FAP is prominently expressed in most soft-tissue sarcomas. Notably, its expression is not limited to fibroblasts; it is also found in other cell types, including macrophages, melanocytes, melanoma, and epithelial tumor cells, highlighting its diverse biological role. In contrast, benign epithelial tumors, normal fibroblasts, and epithelial cells typically do not express FAP. Given its widespread presence across cancer types, FAP has become a significant target for cancer imaging and therapy.
This paper describes the development and validation of an Investigational Medicinal Product Dossier (IMPD) for [⁶⁸Ga]Ga-FAPI-46, prepared in two centers using different synthesis platforms. This work is relevant given the clinical importance of FAP-targeted imaging agents, and the authors provide a comprehensive description of synthesis, quality control, and validation procedures in compliance with current regulatory standards.
The manuscript can be considered for acceptance after addressing the following points:
Major Comments
- Ensure that all figures are provided in high resolution, with detailed captions describing the parameters. For the chromatographic data, redraw the radio-HPLC and HPLC profiles using appropriate software to improve resolution.
- Include the radio-TLC data, which is mentioned in the methods but not shown in the results.
- Tables 6 and 7 are too long; condensing the data would improve readability and presentation.
- The Discussion section is too brief. Please expand it to compare [⁶⁸Ga]Ga-FAPI-46 with other FAPI derivatives in terms of synthesis, stability, and diagnostic performance.
Minor Comments
- Abbreviations (e.g., IMP, IMPD, Ph. Eur.) should be defined at first mention only and used consistently throughout the text.
- Define gallium-68 as ⁶⁸Ga and use this consistently throughout the manuscript.
- Ensure consistent terminology (e.g., use either “tumor” or “tumour” consistently).
- In Figure 1 (chemical structure), please clarify what the symbol “x” represents.
- In Figure 2, the title should be corrected to “⁶⁸Ga decay scheme” rather than “gallium decay scheme.”
Author Response
Dear Reviewer,
We would like to reassure you that we have improved the quality of the images included in the manuscript.
The TLC analysis data can be found in rows 8 and 9 of Tables 6 and 7, respectively.
While it is true that Tables 6 and 7 are lengthy, they contain essential data for anyone looking to replicate the IMPD development model. We believe that omitting some of this data in order to shorten the tables would result in significant information being missing for anyone reading our article who wishes to follow our IMPD development model.
As requested, we have added a discussion section on the synthesis, stability, and diagnostic performance characteristics of [68Ga]Ga-FAPI-46. As this is not the focus of our article, we have not discussed it at length, so as not to distract from the main objective of our manuscript, which focuses on the descriptive aspects of the IMPD development model for a multicentre study using a diagnostic radiopharmaceutical with reduced stability. However, we appreciate your suggestion and hope that our additions to this section address it.
I can confirm that all minor revisions have been made to the article, except for the replacement of 'gallium-68' with '68Ga', which aligns with the January 2024 EANM guidelines, 'A Practical Short Guidance on Correct Radiotracer Nomenclature in Nuclear Medicine'.
Thank you for taking the time to review our article.
Round 2
Reviewer 3 Report
Comments and Suggestions for Authors
The authors have adequately revised the manuscript and addressed the comments. The manuscript is now clear, well-structured, and suitable for publication in Pharmaceuticals.